# Facile Fabrication of Oxygen-Releasing Tannylated Calcium Peroxide Nanoparticles

**DOI:** 10.3390/ma13173864

**Published:** 2020-09-01

**Authors:** Ji Sun Park, Yeong Jun Song, Yong Geun Lim, Kyeongsoon Park

**Affiliations:** Department of Systems Biotechnology, Chung-Ang University, Anseong, Gyeonggi 17546, Korea; park41917235@gmail.com (J.S.P.); rhksdn9502@naver.com (Y.J.S.); kgus0113@naver.com (Y.G.L.)

**Keywords:** calcium peroxide, tannic acid, oxygen generation

## Abstract

This study reports a new approach for the facile fabrication of calcium peroxide (CaO_2_) nanoparticles using tannic acid (TA) as the coordinate bridge between calcium ions. Tannylated-CaO_2_ (TA/CaO_2_) nanoparticles were prepared by reacting calcium chloride (CaCl_2_) with hydrogen peroxide (H_2_O_2_) in ethanol containing ammonia and different amounts of TA (10, 25, and 50 mg). The prepared TA/CaO_2_ aggregates consisted of nanoparticles 25–31 nm in size. The nanoparticles prepared using 10 mg of TA in the precursor solution exhibited the highest efficiency for oxygen generation. Moreover, the oxygen generation from TA (10 mg)/CaO_2_ nanoparticles was higher in an acidic environment.

## 1. Introduction

The delivery of sufficient amounts of oxygen to transplanted cells and 3D-engineered tissues remains among the main challenges in the field of cellular engineering [1]. Oxygen deficiency in these cells results in poor extracellular matrix deposition, cell death, and tissue necrosis [2]. Moreover, hypoxic tumor cells in solid tumor tissues are more resistant to radiation and chemotherapy [3,4]. Therefore, the prevention of hypoxia in 3D scaffolds during transplantation and in solid tumors during cancer therapy is essential for enhancing therapeutic effects.

Recent studies have focused on designing materials to supply oxygen to hypoxic regions [1,5,6]. Among the inorganic peroxide oxygen-releasing compounds such as calcium peroxide (CaO_2_), magnesium peroxide (MgO_2_), and sodium percarbonate (Na_2_CO_3_·1.5H_2_O_2_) [1,5], CaO_2_ is preferred, owing to its higher oxygen-generation capacity [7].

Several methods have been reported for the preparation of CaO_2_ particles by reacting Ca(OH)_2_ with H_2_O_2_ in an aqueous solution [8] or reacting CaSO_4_ with H_2_O_2_ in aqueous KOH [9]. However, the CaO_2_ particles formed in aqueous solutions tend to aggregate because of hydrolysis. To overcome this, polyethylene glycol has been used as a surface modifier to stabilize the CaO_2_ nanoparticles [10,11]. However, this method involves a time-consuming washing process to stabilize the pH. Moreover, the resultant products are non-uniform in size. To address these drawbacks, Shen et al. reported an alternative method for the preparation of CaO_2_ nanoparticles using polyvinyl pyrrolidine as a colloidal stabilizer [12].

Tannic acid (TA), a polyphenolic biomolecule, is a versatile surface modifier due to the presence of phenolic hydroxyl groups, aromatic rings, and galloyl groups [13,14]. In addition, the tendency of TA to coordinate with metal ions has been exploited to synthesize Ag and Au nanoparticles [15,16]. Most importantly, additional coordinate bridges are formed between the catechol moieties of TA and calcium ions of Ca-alginate nanoparticles [17]. Based on these observations, the present study reports a new approach for the facile fabrication of CaO_2_ nanoparticles using TA as a coordinate bridge between calcium ions. Additionally, the oxygen-releasing capability of the prepared CaO_2_ nanoparticles is examined.

## 2. Materials and Methods

### 2.1. Materials

Calcium chloride dihydrate (CaCl_2_·2H_2_O, abbreviated as CaCl_2_ hereafter), TA, ammonia solution (NH_4_OH, 1 M), hydrogen peroxide solution (H_2_O_2_, 35 wt%), and potassium bromide (KBr) were purchased from Sigma-Aldrich (St. Louis, MO, USA). Phosphate-buffered saline (PBS) was obtained from Lonza (Walkersville, MD, USA). Absolute ethanol (EtOH, 99.9%), methanol (MeOH, 99%), and isopropanol were purchased from DUKSAN (Ansan, Korea).

### 2.2. Preparation of Tannylated Calcium Peroxide (TA/CaO_2_) Nanoparticles

CaCl_2_ (100 mg) and TA (10, 25, and 50 mg) were dissolved in 15 mL of EtOH or MeOH using a sonicator (BRANSON 5510R-MT, Marshall Scientific, Hampton, NH, USA). Under constant stirring at 600 rpm, 0.8 mL of NH_4_OH (1 M) was added to this mixture and reacted for 2 min. Following this, 0.16 mL of H_2_O_2_ (35 wt%) was added dropwise at a rate of 0.05 mL/min using a syringe pump (NE-1000, New Era Pump Systems Inc., Farmingdale, NY, USA). After sonication for 10 min, the mixtures were left to react overnight. The TA (10, 25, or 50 mg)/CaO_2_ nanoparticles were subsequently obtained by centrifugation of the mixtures at 28,620 g for 10 min. All these synthetic procedures were performed at room temperature. The powders were washed with EtOH thrice and dispersed in 16 mL of EtOH for storage.

### 2.3. Characterization of TA/CaO_2_ Nanoparticles

To determine the mean yield of TA/CaO_2_, a rotary evaporator (N-1200BS, EYELA, Bohemia, NY, USA) was used to completely remove EtOH from the samples. The TA/CaO_2_ powder thus obtained was characterized by Fourier transform infrared (FT-IR, Shimadzu 8400S, Kyoto, Japan) spectroscopy. The FT-IR spectrum was acquired using the KBr pellet method at a resolution of 4 cm^–1^ between 4000 and 400 cm^–1^.

The particle size, polydispersity index, and Z-average size of each sample (2 mL) dispersed in EtOH were measured at a scattering angle of 90° with a particle size analyzer (SZ-100, HORIBA, Kyoto, Japan). Frequency (%) of particle size data means the amount of each size by volume. For scanning electron microscopy (SEM) analysis, EtOH of each TA/CaO_2_ sample was removed using a rotary evaporation at 30 °C for 1 h, then further dried in a drying oven at 60 °C for 1 day to remove EtOH. After this, the dried TA/CaO_2_ powders were coated with platinum, then their morphologies, sizes, and elemental compositions were investigated by field-emission SEM (FE-SEM, S-4700, Hitachi, Chiyoda, Japan). The average size of the randomly selected individual nanoparticles (n > 60) in each TA/CaO_2_ sample were analyzed using Image J software (Version 1.47; US National Institutes of Health, Bethesda, MD, USA). The elemental compositions of the samples were determined by energy-dispersive X-ray spectroscopy (EDS). The thermal degradation of the samples was evaluated by thermal gravimetric analysis (TGA) (TA Q600, TA instrument, New Castle, PA, USA). The samples were heated from 40 to 700 °C at a rate of 10 °C/min.

### 2.4. Visualization of O_2_ Bubble Formation and Dissolved Oxygen Measurement

We added 300 μL of TA (10 mg)/CaO_2_ sample to 2 mL of each of the PBS solutions (pH 6.0 and 7.4) at room temperature. The O_2_ bubbles thus generated were monitored for 10 min using a smartphone camera (iPhoneXR, Apple, Cupertino, CA, USA).

The concentration of dissolved oxygen was recorded for 10 min using an Oakton DO 6+ dissolved oxygen meter (EW-35643-15, Cole-Parmer, Vernon Hills, IL, USA) after adding 3 mL of TA (10 mg)/CaO_2_ to 20 mL N_2_-purged PBS solutions (pH 6.0 and 7.4).

### 2.5. Cytotoxicity Test

Cytotoxicity test of the dried TA (10 mg)/CaO_2_ against macrophages (RAW 264.7, Korean Cell Line Bank, Seoul, Korea) was investigated with a cell-counting kit-8 (CCK-8; Dojindo Molecular Technologies, Inc., Gumamoto, Japan). Cells were cultured and maintained in Dulbecco’s modified Eagle’s medium (Welgene, Gyeongsan, Korea) containing 10% fetal bovine serum and 1% penicillin-streptomycin using a humidified 5% CO_2_ incubator. For the cytotoxicity test, macrophages (1 × 10^4^ cells/well) were seeded into a 96-well culture plate and incubated for 24 h. Then, the cells were cultured in nitrogen (N_2_)-purged culture medium containing various concentrations of TA (10 mg)/CaO_2_ (0, 25, 50, and 100 μg/mL) at 37 °C for 24 h, after which the cells were washed with fresh media and CCK-8 reagent solution was treated to the cells. The cells were further incubated for 2 h, and the optical density was determined at 450 nm using a microplate reader (Fisher Scientific, Hampton, NH, USA).

### 2.6. Statistical Analysis

The data represented mean ± standard deviation, and mean values were compared between two groups by *t*-test or between many groups by one-way ANOVA (Systat Software, Inc., Chicago, IL, USA). *p* values less than 0.05 were considered statistically significant.

## 3. Results and Discussion

### 3.1. Preparation and Characterization of TA/CaO_2_ Nanoparticles

TA/CaO_2_ nanoparticles were prepared by the reaction of calcium chloride (CaCl_2_) solution with H_2_O_2_ in EtOH containing ammonia solution and different amounts of TA (10, 25, and 50 mg) forming coordinate bridges with calcium ions (Equation (1), Figure 1). CaO_2_ nanoparticles were initially formed after adding H_2_O_2_ into the EtOH solution containing CaCl_2_ and TA. Through the coordination of TA with calcium ions, small-sized TA/CaO_2_ particles were formed at an initial reaction time and larger aggregates were subsequently obtained. The experimental yields of TA (10 mg)/CaO_2_, TA (25 mg)/CaO_2_, and TA (50 mg)/CaO_2_ were 43.20 ± 2.25 mg, 61.97 ± 7.80 mg, and 83.7 ± 3.24 mg, respectively, and the yield amounts increased with increasing TA amounts.
CaCl_2_ + H_2_O_2_ + 2NH_4_OH → CaO_2_ + 2NH_4_Cl + 2H_2_O,(1)

The formation of TA/CaO_2_ was confirmed with FT-IR analysis. Appendix A shows a wide and strong absorption band centered at 3380 cm^–1^, which is assigned to the –OH stretching mode resulting from hydrogen bonding. Owing to the presence of ester groups, peaks were detected at 1640 and 1100 cm^–1^, arising from the C=O and C–O stretching modes, respectively. The characteristic peaks at 1492 and 1420 cm^–1^ are associated with the bending mode of O–Ca–O [18].

The observed mean sizes of three TA/CaO_2_ aggregates ranged from 220 to 263 nm, and their Z-average sizes ranged from 200 to 240 nm in diameter, with the midrange polydispersity indexes (PDIs) ranging from 0.187 to 0.383 (Figure 2a,c and Appendix A). Noticeable dependency for the particle sizes was not observed. Also, when three TA/CaO_2_ aggregates were stored in EtOH at room temperature without any agitation, they began to be slightly aggregated after two days storage (data not shown). SEM images show that all three TA/CaO_2_ samples exhibited spherical shapes (Figure 2d,f). Interestingly, the individual nanoparticle sizes were approximately 26.94 ± 5.29 nm for TA (10 mg)/CaO_2_, 31.29 ± 6.18 nm for TA (25 mg)/CaO_2_, and 25.21 ± 4.06 nm for TA (50 mg)/CaO_2_. The particle sizes of three TA/CaO_2_ nanoparticles determined by dynamic light scattering appeared to be much larger than those measured by SEM in a dried state because the interactions between individual particles in EtOH might facilitate their aggregation to form much larger particles.

Through SEM-EDS analysis, we further confirmed the presence of TA/CaO_2_. As shown in Figure 3a,c and Appendix A, there was a significant increase in C content and a decrease in Ca content as the amount of TA was increased from 10 to 50 mg. In addition, the EDS maps confirmed the presence of C, O, and Ca elements in the nanoparticles. TGA revealed that the thermal degradation rates of the TA/CaO_2_ nanoparticles increased as the amount of TA was increased from 10 to 50 mg (Appendix A). Thus, the results confirmed the successful synthesis of TA/CaO_2_ nanoparticles.
2CaO_2_ + 4H_2_O → 2Ca(OH)_2_ + 2H_2_O_2_ → 2Ca(OH)_2_ + 2H_2_O + O_2_,(2)

### 3.2. Confirmation of Oxygen Generation

CaO_2_ hydrolyzes to generate H_2_O_2_ and, subsequently, O_2_, when in contact with water, as shown in Equation (2) [12]. To determine the oxygen generation efficiency of TA/CaO_2_ by hydrolysis, a fixed amount of each TA/CaO_2_ sample was added to PBS (pH 7.4) and the amount of dissolved oxygen in the solution was measured (Figure 4). In PBS (pH 7.4), the amount of dissolved oxygen decreased with increasing TA in the precursor. This result indicates that the oxygen generation efficiency of the TA/CaO_2_ nanoparticles is affected by the production of H_2_O_2_ by hydrolysis. Based on this result, the TA (10 mg)/CaO_2_ sample was selected for further experiments.

It has been reported that solid CaO_2_ is relatively stable against decomposition but is soluble in acids [19]. This is consistent with our results, where TA/CaO_2_ powder was more soluble in pH 6.0 than in pH 7.4 (Appendix A). To further examine whether the oxygen generation efficiency of TA (10 mg)/CaO_2_ dispersed in EtOH is enhanced in an acidic environment, the generation of oxygen bubbles was monitored for 10 min following the addition of the sample in PBS (pH 6.0). The dissolved oxygen concentration was also measured in pH 6.0 and pH 7.4 PBS solutions. Figure 5 shows that TA (10 mg)/CaO_2_ generates more oxygen bubbles in pH 6.0 than in pH 7.4. Moreover, the oxygen generated from TA (10 mg)/CaO_2_ maintained a higher concentration in acidic pH 6.0 than in pH 7.4.

### 3.3. Preparation, Particle Sizes and Oxygen Generation of TA/CaO_2_ Nanoparticles in Other Alcoholic Solvents

Typically, TA/CaO_2_ nanoparticles are synthesized in EtOH. To further investigate whether TA/CaO_2_ nanoparticles could be synthesized in other alcoholic solvents, we first tested the solubility of TA in MeOH, EtOH, and isopropanol. Appendix A shows that TA is soluble in MeOH and EtOH, but insoluble in isopropanol, indicating that MeOH can be also used as a solvent for synthesizing TA (10 mg)/CaO_2_ nanoparticles. The mean sizes of TA (10 mg)/CaO_2_ nanoparticles prepared in MeOH were 236.6 ± 57.8 nm and their Z-average size was 216.5 nm in diameter with 0.209 of PDI (Appendix A). Also, they could generate oxygen bubbles in pH 7.4 PBS. These data suggest that TA/CaO_2_ nanoparticles could be also prepared in MeOH and have oxygen generation potency.

### 3.4. Cytotoxicity Test

Cytotoxicity study of TA (10 mg)/CaO_2_ towards macrophages was performed using the CCK-8 assay. Compared to control group, the cell viability increased with increasing the concentration of TA (10 mg)/CaO_2_ (Figure 6). In particular, the cell viability treated with 100 μg/mL of TA (10 mg)/CaO_2_ was much higher than that of control group. This increased cell-viability may be related to the oxygen generation from of TA (10 mg)/CaO_2_, and the oxygen generation may influence cell viability of macrophages.

## 4. Conclusions

TA/CaO_2_ nanoparticles were synthesized using EtOH as the solvent and TA as the coordinate bridge with calcium ions. The spherical aggregates consisted of nanoparticles with sizes in the range of 25–31 nm. Among the three samples synthesized with different amounts of TA in the precursor, TA (10 mg)/CaO_2_ generated H_2_O_2_ and O_2_ most efficiently upon contact with water. Also, TA (10 mg)/CaO_2_ nanoparticles released more oxygen bubbles in an acidic environment. Furthermore, TA (10 mg)/CaO_2_ nanoparticles may improve cell viability followed by oxygen generation.

## Figures and Tables

**Figure 1 materials-13-03864-f001:**
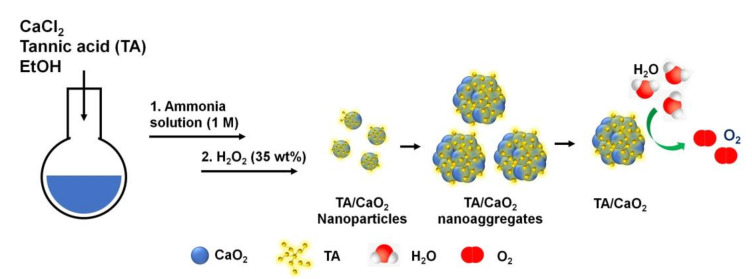
Schematic illustration of the synthesis of tannylated calcium peroxide (TA/CaO_2_) nanoparticles and oxygen generation from TA/CaO_2_ nanoparticles as a result of hydrolysis.

**Figure 2 materials-13-03864-f002:**
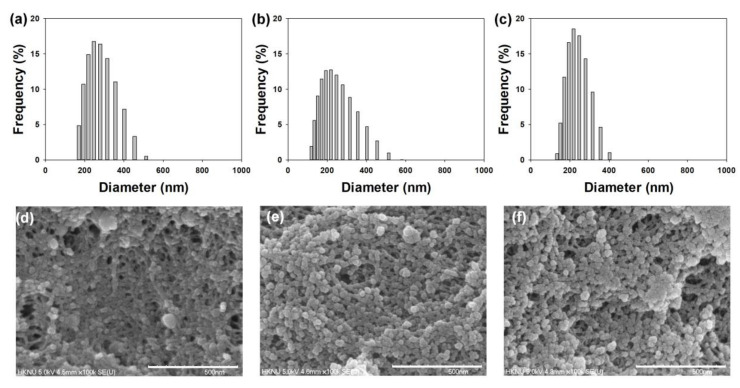
Particle size distributions of (**a**) TA (10 mg)/CaO_2_, (**b**) TA (25 mg)/CaO_2_, and (**c**) TA (50 mg)/CaO_2_. SEM images of (**d**) TA (10 mg)/CaO_2_, (**e**) TA (25 mg)/CaO_2_, and (**f**) TA (50 mg)/CaO_2_. Scale bar: 500 nm.

**Figure 3 materials-13-03864-f003:**
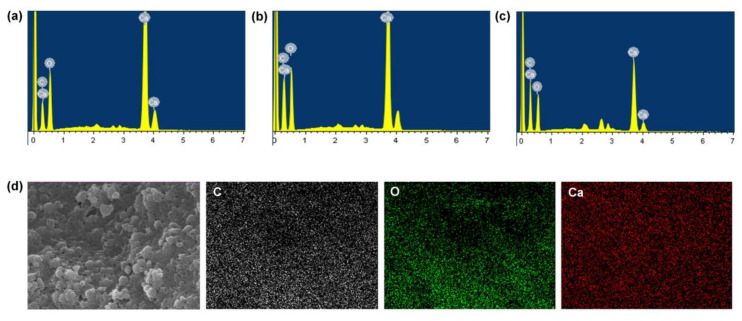
SEM-EDS results of (**a**) TA (10 mg)/CaO_2_, (**b**) TA (25 mg)/CaO_2_, and (**c**) TA (50 mg)/CaO_2_. (**d**) SEM image and the corresponding EDS elemental maps of carbon (C; white), oxygen (O; green), and calcium (Ca; red) in TA (10 mg)/CaO_2_ nanoparticles.

**Figure 4 materials-13-03864-f004:**
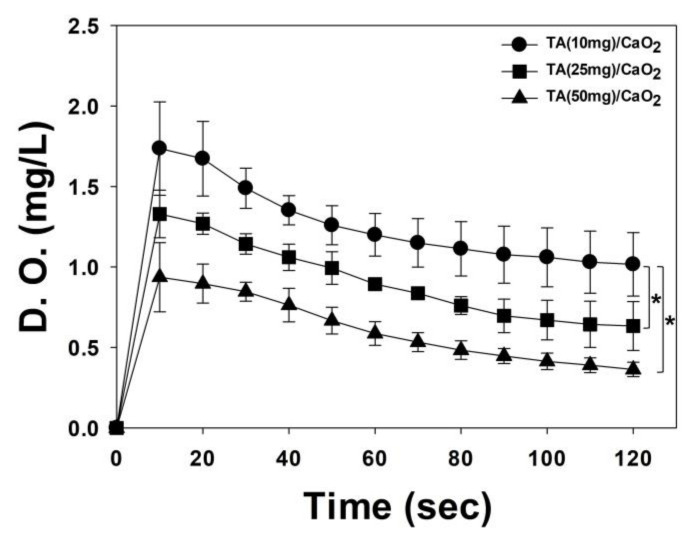
Oxygen concentration in suspensions of TA (10 mg)/CaO_2_, TA (25 mg)/CaO_2_, and TA (50 mg)/CaO_2_ nanoparticles, as measured using a portable dissolved oxygen meter. The data represent mean ± SD (n = 3). * *p* < 0.05.

**Figure 5 materials-13-03864-f005:**
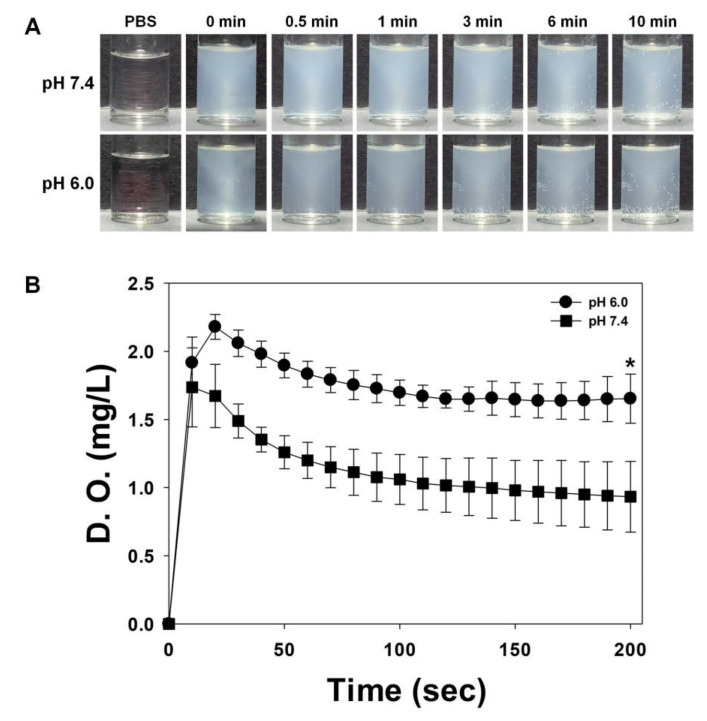
(**A**) Representative photographs of oxygen bubbles generated upon hydrolysis of TA (10 mg)/CaO_2_ at pH 7.4 and 6.0. (**B**) Oxygen concentrations in suspensions of the TA (10 mg)/CaO_2_ nanoparticles, as measured using a portable dissolved oxygen meter. The data represent mean ± SD (n = 3). * *p* < 0.05.

**Figure 6 materials-13-03864-f006:**
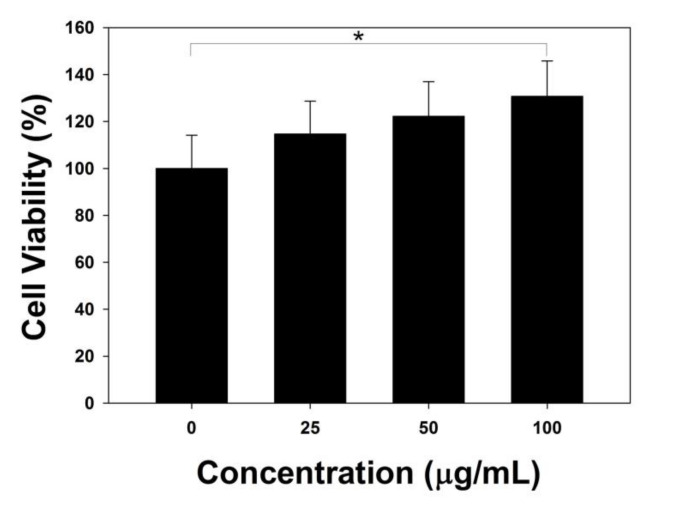
Cytotoxicity test of TA (10 mg)/CaO_2_ towards macrophages. The data represent mean ± SD (n = 5). * *p* < 0.05.

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
