# Peer review of "Facile Fabrication of Oxygen-Releasing Tannylated Calcium Peroxide Nanoparticles"

_materials, 2020, doi:10.3390/ma13173864_

Round 1

Reviewer 1 Report

The authors in this article present an interesting alternative for the preparation of calcium peroxide nanoparticles. The experimental design is sound, although there are some questions raised i my opinion. More specifically:

1) The PdI of TA(10mg)/CaOand TA(25mg)/CaO2 are relatively high, which as the authors state indicate the tendency to aggregate. Were any stability studies performed? 

2) The generation of hydrogen peroxide study by CaO2 was performed at the same quantities for all three tannic acid contents. Since the elemental analysis showed the decrease in calcium content as tannic acid content increased, wouldn't it be more accurate to compare the same molar quantities? As it is, attributing the absence of chemiluminescence at tannic acid's  scavenging activity is not accurate, as different amounts of calcium peroxide are being compared.

Author Response

We submitted responses to reviewer's comments as upload word file.

Reviewer 2 Report

Review comments are attached

Author Response

(The authors gave the same response as above.)

Reviewer 3 Report

Paper very well organized, well written, objective and concise, which makes it rigorous and easy to analyze. Congratulations. The subject is very relevant, pertinent and current. Please consider my suggestions as possible improvements.

The title starts with the word "Fabile" yet in the abstract you discuss "facile fabrication". This might be a typo.

The section for the statistical treatment used is missing.

Line 154-155: At this point of the discussion, is the utility of this compound for physiological use maintained, as was discussed in the introduction?

There should be a control for comparing with compounds already known for this same purpose.

Line 159: It would be important to see the behaviour of an already known compound that might serve as control in this analysis. Including in the discussion the comparison between the behaviour of both compounds in the conditions tested.

Line 166-167: Please consider reformulating this sentence and remove the personalisation (the paper should be written in the third person so "we" or "our" must not be used in the manuscript and should be rephrased such as "it was presented..., it was examined..., the data showed..."). Given that no testing was done on cancer cells or tissues, the conclusion as it is written cannot be made.

Author Response

(The authors gave the same response as above.)

Reviewer 4 Report

Authors reported a "new approach" for the facile fabrication of calcium peroxide (CaO2) nanoparticles using tannic acid (TA) as the coordinate bridge between calcium ions. The novelty of this preparation method was not clearly demonstrated. Several methods already described in the literature including tannic acid. Authors must clearly described the differences between them.

Cell studies are strongly recommended, confirming the aim of those nanoparticles.

Statistical analysis is missing.

It is not completely understood the differences in terms of DLS and SEM.

Size and zeta analysis is incomplete (solvent, temperature, angle, dilution, frequency in what?). Same for SEM analysis.

Centrifugation units must be done using g and not rpm. Experimental conditions such as temperature must be clearly described.

Missing SD in Figure 4. Same for figure 5.

Toxicological behaviour of those particles should be performed.

Some gramatical errors along the manuscript (e.g. line 61).

Author Response

(The authors gave the same response as above.)

Round 2

Reviewer 2 Report

Review comments are attached

Author Response

We already submitted the responses to reviewer-2' comments. We attached the file regarding the response to reviewer-2's comments. Please see the attached file and review. 

Reviewer 4 Report

Most of the queries were answered by authors. The quality of the paper was improved. However, cell studies must be done because this is the aim of this study. I encourage to resubmit this paper after inclusion of cell studies. 

Author Response

We attached the file regarding the response to reviewer's comments. See the attached file. 

Round 3

Reviewer 4 Report

Cell studies were performed.